# Probing the link between vision and language in material perception using psychophysics and unsupervised learning

**Chenxi Liao** [1] *, **Masataka Sawayama** [2], **Bei Xiao** [3]

**1** American University, Department of Neuroscience, Washington DC, United States of America, **2** The University of Tokyo, Graduate School of Information Science and Technology, Tokyo, Japan, **3** American University, Department of Computer Science, Washington DC, United States of America

* cl6070a@american.edu

**Data Availability Statement:** Human and model data, data analysis code, stimuli, training data, and trained networks are available on Github (https:// github.com/cl3789/Material-Morphing-toolkit-git)

## Abstract

We can visually discriminate and recognize a wide range of materials. Meanwhile, we use language to describe what we see and communicate relevant information about the materials. Here, we investigate the relationship between visual judgment and language expression to understand how visual features relate to semantic representations in human cognition. We use deep generative models to generate images of realistic materials. Interpolating between the generative models enables us to systematically create material appearances in both well-defined and ambiguous categories. Using these stimuli, we compared the representations of materials from two behavioral tasks: visual material similarity judgments and free-form verbal descriptions. Our findings reveal a moderate but significant correlation between vision and language on a categorical level. However, analyzing the representations with an unsupervised alignment method, we discover structural differences that arise at the image-to-image level, especially among ambiguous materials morphed between known categories. Moreover, visual judgments exhibit more individual differences compared to verbal descriptions. Our results show that while verbal descriptions capture material qualities on the coarse level, they may not fully convey the visual nuances of material appearances. Analyzing the image representation of materials obtained from various pre-trained deep neural networks, we find that similarity structures in human visual judgments align more closely with those of the vision-language models than purely vision-based models. Our work illustrates the need to consider the vision-language relationship in building a comprehensive model for material perception. Moreover, we propose a novel framework for evaluating the alignment and misalignment between representations from different modalities, leveraging information from human behaviors and computational models.

## Author summary

Materials are building blocks of our environment, granting access to a wide array of visual experiences. The immense diversity, complexity, and versatility of materials present challenges in verbal articulation. To what extent can words convey the richness of visual

and Figshare (https://figshare.com/articles/dataset/
Probing_the_Link_between_Vision_and_
Language_in_Material_Perception/25134380 and
https://figshare.com/articles/dataset/Space_of_
Morphable_Material_Appearance_Datasets_and_
Models/27038518).

**Funding:** This study was supported by the National
Institutes of Health to BX, 1R15EY033512-01A1.
The funders had no role in study design, data
collection and analysis, decision to publish, or
preparation of the manuscript. BX received
summer salaries from NIH via the award
(1R15EY033512-01A1).

**Competing interests:** The authors have declared
that no competing interests exist.

material perception? What are the salient attributes for communicating about materials?
We address these questions by measuring both visual material similarity judgments and
free-form verbal descriptions. We use AI models to create a diverse array of plausible
visual appearances of familiar and unfamiliar materials. Our findings reveal a moderate
vision-language correlation within individual participants, yet a notable discrepancy per-
sists between the two modalities. While verbal descriptions capture material qualities at a
coarse categorical level, precise alignment between vision and language at the individual
stimulus level is still lacking. These results highlight that visual representations of materi-
als are richer than verbalized semantic features, underscoring the differential roles of lan-
guage and vision in perception. Lastly, we discover that deep neural networks pre-trained
on large-scale datasets can predict human visual similarities at a coarse level, suggesting
the general visual representations learned by these networks carry perceptually relevant
information for material-relevant tasks.

## Introduction

We often describe what we see with words. Language reveals how we interpret and communi-
cate our sensory experiences and provides critical information about our mental representa-
tion of the environment [1]. The interaction between language and perception has long been
debated, mainly in visual cognition, such as color categorization [2–4] and scene interpretation
[5, 6]. Jointly modeling visual and natural language features expands the capability of artificial
intelligence systems (e.g., image-classification [7, 8], image-retrieval [9–11], and text-to-image
generation [12, 13]) and provides valuable tools for investigating the neural correlates of object
and scene recognition [14, 15]. Little is known about how and what aspects we communicate
about materials, which are the building blocks of objects and the environment. Material per-
ception facilitates us to form a vivid and rich representation of the external world, which in
turn guides our interaction with it. Although we can visually recognize and discriminate a
broad range of materials, we might find it challenging to precisely and effectively describe
their appearances and properties with words. To what extent do words encapsulate the rich-
ness of visual material perception? What are the salient attributes for communicating about
materials?

Based on visual input, we can often distinguish materials, and infer their diverse optical
properties (e.g., surface glossiness [16–19], translucency [20–29] or transparency [30]), surface
properties (e.g., roughness [31]), mechanical properties (e.g., softness [32], stiffness [33]) and
states (e.g., freshness [34], wetness [35]). Previous works actively examined how visual esti-
mates of material attributes are related to the statistical image features [36], as well as seeking
to probe the neural representation of material perception in cortical areas of the ventral visual
pathway [37–40]. Along with visual discrimination, verbalizing what we see reflects, to a cer-
tain degree, how we process and organize visual information into semantic-level representa-
tion. Verbal description could serve as an interpretable representation that encodes the salient
features of material qualities. While a plethora of works scrutinized the visual estimation of
specific material properties related to physics [17, 36, 41–43], few studies shined the light on
more subjective material perception from both visual judgment and language expression.
With a broad dataset of natural material images, Schmidt et al. (2022) [44] used visual triplet
similarity judgments from crowd-sourcing to distill a representational space, which was later
annotated by humans to find conceptual and perceptual dimensions of materials. Cavdan et al.
(2023) [45] studied the structure of the representational space of perceptual softness triggered

by material name with a cross-group analysis and suggested that verbally activated softness representation correlates with that derived from vision [32]. However, participants in these studies were often limited to judging materials based on predetermined categories and attributes without being given the opportunity to express their personal semantic interpretations. Further, previous works typically focused on the group-level analysis and downplayed the potential individual variances. To definitively assess the link between vision and language in material perception, it is crucial to measure visual judgment and verbal description within the individual participants, as well as allow them to freely articulate their unique visual experiences.

Perceiving materials may entail processing visual information at multiple levels. To probe the representational space of materials, designing stimuli that incorporate both the naturalness of mid-to-low-level visual properties and semantic-level richness is essential. To achieve this, we developed an effective approach to create an extensive range of plausible visual appearances of familiar and novel materials (see Fig 1). Without requiring annotated data, unsupervised learning has been recently applied to the study of human material perception [29, 46, 47]. We use an unsupervised image synthesis model, StyleGAN2-ADA [48], to generate images of diverse materials based on the learning of real-world photos. As a result, the model parameterizes the statistical structures of material appearances and facilitates linear interpolation between image data points, allowing us to morph between different material categories (e.g., morphing between a soap to a rock results in an ambiguous translucent object shown in Fig 2C). This approach enables us to continuously vary the multidimensional structural features of materials (e.g., the combination of shape and color variation) and build an expanded Space of Morphable Material Appearance. The morphed materials resemble the visual characteristics of both original materials (e.g., soap and rock), potentially resulting in the ambiguity of perceived material identity. This offers an opportunity to investigate the influence of semantic-level interpretation on material perception.

We measured material perception with two behavior tasks involving vision and language within individuals: Multiple Arrangement and Verbal Description (Fig 3). Stimuli were sampled from the Space of Morphable Material Appearance (Fig 2D and 2E). In the Multiple Arrangement task, participants arranged materials based on visual similarities [49]. For the verbal description task, the participants described the same images with texts. With the recent advancements in Large Language Models (LLMs), it is now possible to create a representation based on verbal reports provided by the participants. We discovered a moderate vision-language correlation within individual participants by quantitatively comparing the behavioral representations derived from two tasks. Incorporating two behavioral tasks allows us to uncover salient semantic features associated with material perception. Crucially, by assessing the stimulus-level representational differences, we noted a persistent gap in using words to capture the nuanced visual differences among diverse material samples, particularly among the ambiguous materials generated from morphing.

How do we discover perceptually relevant features that are descriptive of the richness of human material perception? We confronted various pre-trained, large-scale deep neural network models with our stimuli and compared image-based representations extracted from these models with those from humans. Despite not being explicitly trained with our stimuli, we found that these models can cluster images in patterns similar to human perception. Furthermore, we discovered that the weakly-supervised model (e.g., Contrastive Language-Image Pretraining (CLIP) [7] and OpenCLIP [50]), which learns high-level semantic correlations between images and text, best predicts human visual similarity judgments. We discuss the implications of our results on visual and semantic contribution to material perception.

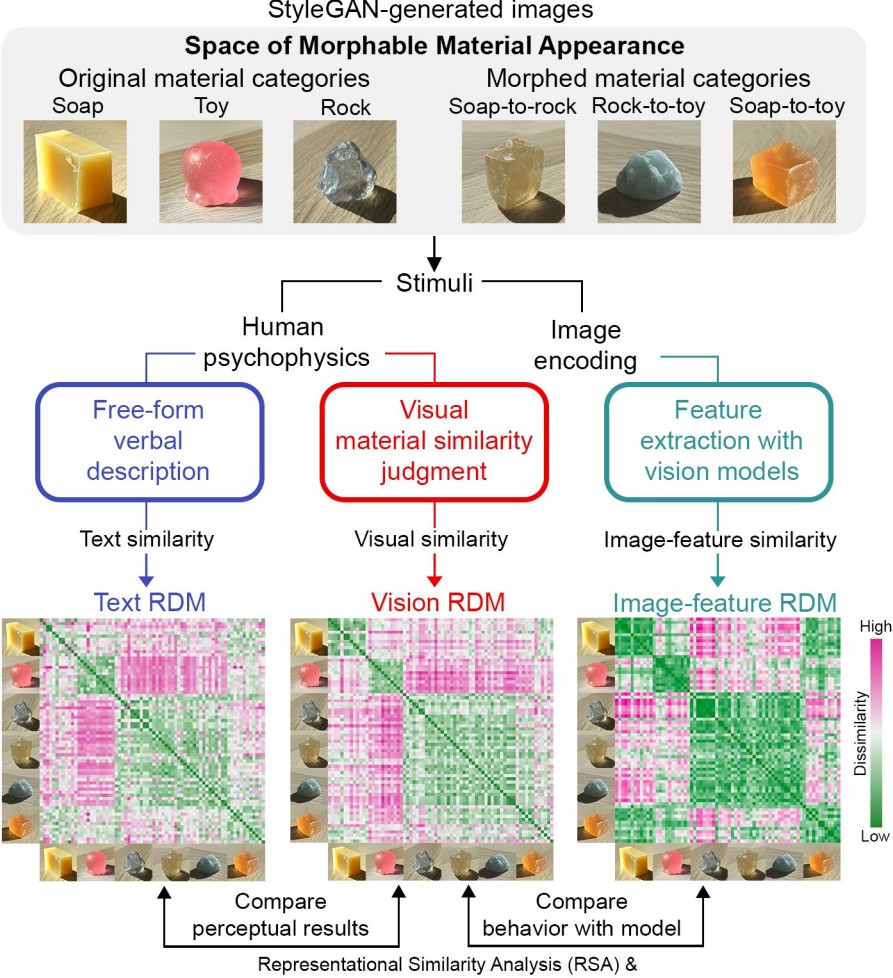

**Fig 1. Our framework to investigate the link between vision and language in material perception.** We built an expandable Space of Morphable Material Appearance based on the unsupervised image generation model StyleGAN (see details in Fig 2). Our method allows us to synthesize images of diverse material appearances in a controllable manner. We created six image categories of material: three original materials (i.e., soap, toy, rock) by directly learning from real photos, and three morphed materials by cross-material morphing (i.e., soap-to-rock, rock-to-toy, and soap-to-toy). The image examples displayed represent each of the six categories. Sampling stimuli from the Space of Morphable Material Appearance, we measured material perception with two psychophysical tasks: visual material similarity judgment and verbal description. We quantitatively evaluated the representations across modalities using the Representational Similarity Analysis (RSA) and the unsupervised alignment method Gromov-Wasserstein Optimal Transport (GWOT). Within each participant, we compared Representational Dissimilarity Matrices (RDMs) between the visual judgment (i.e., Vision RDM) and verbal description (i.e., Text RDM) results. We also compared the participant's visual judgment behavior with the image-feature representations (i.e., Image-feature RDM) of stimuli extracted from the self-supervised (e.g., DINO) or weakly-supervised models (e.g., text-guided model CLIP) pre-trained on large-scale datasets.

## Results

### Space of Morphable Material Appearance

Employing the unsupervised learning model, StyleGAN2-ADA, we generated images of diverse materials with perpetually convincing quality by training on real-world photos (Fig 2). With its multi-scale generative network ($G$) and scale-dependent latent space ($W$), the model learns the

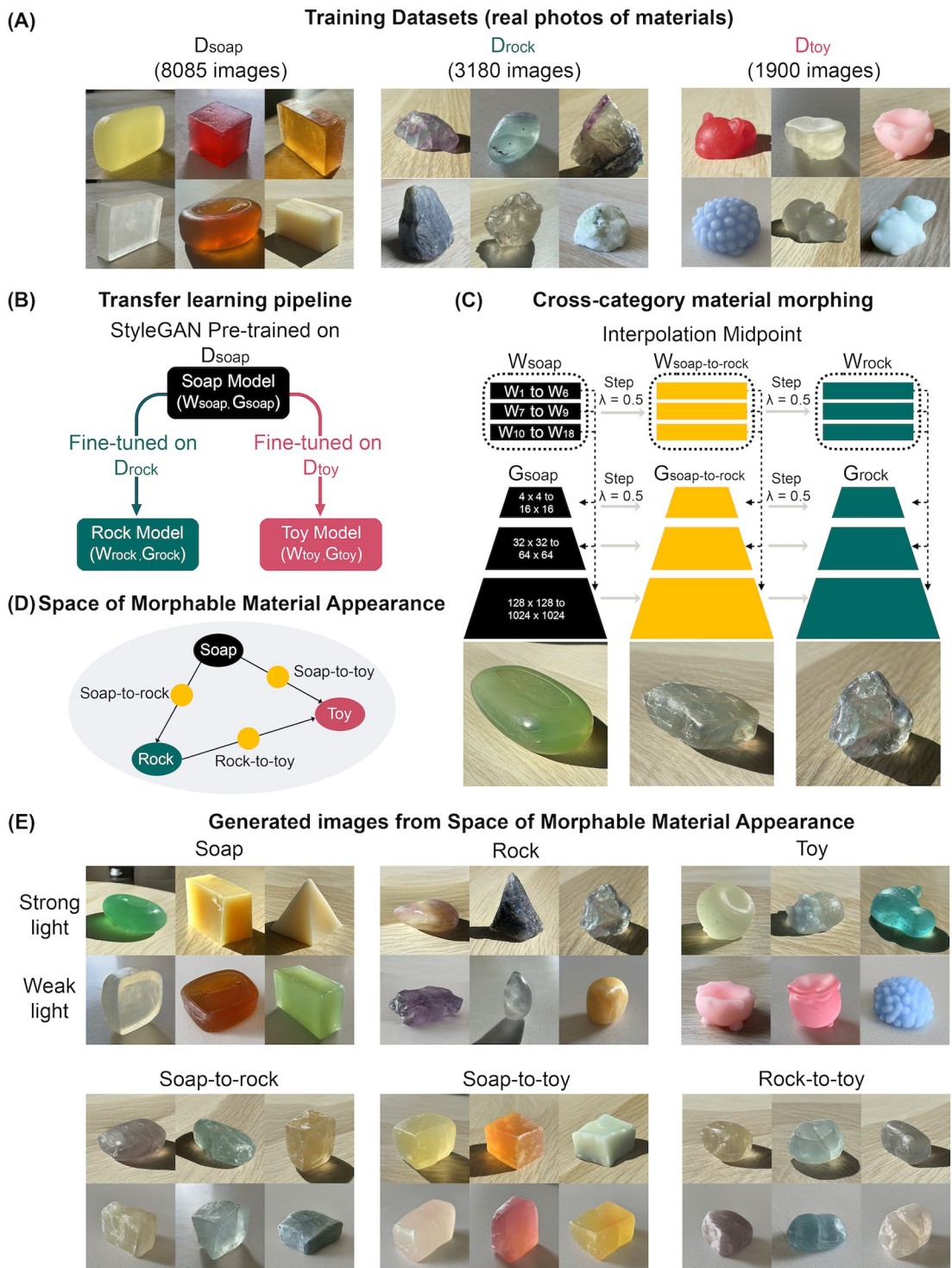

**Fig 2. Overview of the synthesis pipeline for morphable material appearances.** (A) Training datasets. (B) Transfer learning pipeline. Upon training, we obtained models to generate images from three material classes. We can generate images of a desired material (e.g., soaps) by injecting the latent codes (e.g., $w_{soap} \in W_{soap}$) into the corresponding material generator (e.g., $G_{soap}$). (C) Illustration of cross-category material morphing. By linearly interpolating between a soap and a rock, we obtain a morphed material, "soap-to-rock," produced from its latent code $w_{soap-to-rock}$ and generator $G_{soap-to-rock}$. (D) Illustration of the Space of Morphable Material Appearance. (E) Examples of generated images from the Space of Morphable Material Appearance. These images are a subset of stimuli used in our psychophysical experiments, covering two major lighting conditions (i.e., strong and weak lighting).

statistical regularity of the images at multiple spatial scales, spontaneously disentangling semantically meaningful visual attributes, such as the object's shape, texture, and body color [29]. Here, we built our own image datasets that include three materials: soaps ($D_{soap}$), rocks ($D_{rock}$), and squishy toys ($D_{toy}$) (Fig 2A). We fine-tuned the StyleGAN pre-trained on the large soap dataset $D_{soap}$ on the smaller datasets $D_{rock}$ and $D_{toy}$ (Fig 2B). With a short training time, the Soap Model ($W_{soap}$, $G_{soap}$) turned into Rock ($W_{rock}$, $G_{rock}$) and Toy Models ($W_{toy}$, $G_{toy}$) and can synthesize images of realistic and diverse rocks/crystals and squishy toys, under the broad variation of three-dimensional (3D) shapes, colors, textures, and lighting environments (Fig 2E Top Row). The effectiveness of transfer learning also suggests that the different categories of materials have common visual characteristics, such as color variation, specular highlight, and surface geometry; thus, learning features from one material benefits learning new materials.

We can produce novel material appearances without additional training, by morphing between existing learned materials. Given the images of a pair of source and target materials, we can linearly interpolate between their layer-wise latent codes (e.g., $w_{soap}$ and $w_{rock}$) while interpolating all convolution layers' weight parameters of the corresponding material generators (e.g., $G_{soap}$ and $G_{rock}$) (see Method). At a given step size, we can synthesize the image of a morphed material with the interpolated latent code (e.g., $w_{soap-to-rock}$) and generator (e.g., $G_{soap-to-rock}$). Fig 2C illustrates the method of creating a morphed material between soap and rock. In S7 Fig, we further illustrate the cross-material morphing with additional interpolation steps.

Combining transfer learning and model morphing, we constructed an expandable Space of Morphable Material Appearance, from which we can systematically sample and create existing and novel material appearances with object-level realism (Fig 2D). In this study, we focused on the material appearances at the morphing midpoints (i.e., step $\lambda = 0.5$). With this technique, we generated morphed materials, soap-to-rock (midpoint from soap to rock), soap-to-toy (midpoint from soap to squishy toy), and rock-to-toy (midpoint from rock to squishy toy) (Fig 2E Bottom row). We sampled 72 images from the Space of Morphable Material Appearance as stimuli for both of our behavioral experiments (see S1 Fig, S1 Text, and Method).

## Visual material judgment and verbal description are moderately correlated within individuals

Using the above-mentioned stimuli, we measured material perception with Multiple Arrangement and Verbal Description tasks. In the Multiple Arrangement task, participants were

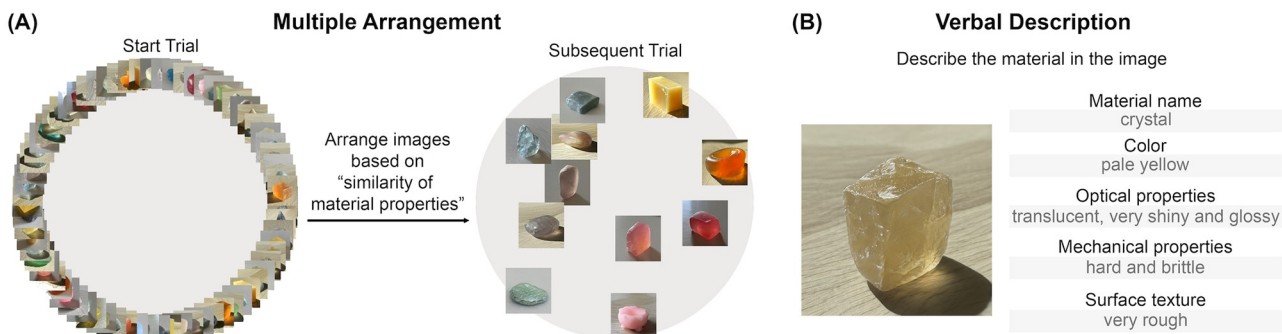

**Fig 3. Illustrations of psychophysical experiment interface.** (A) The Multiple Arrangement task. Participants (N = 16) arranged images within a circle based on their judgment of the visual similarity of material properties. In the first trial, participants were presented with all 72 images of materials. In each subsequent trial, a subset of images was iteratively presented based on an adaptive sampling algorithm [49]. (B) The Verbal Description task. With free-form text input, participants were asked to describe the material shown in the image from five aspects: material name, color, optical properties, mechanical properties, and surface texture. The gray texts are example responses from one trial.

instructed to place the images within the circled region based on the "similarity of material properties" (Fig 3A). The task prompted the consideration of various aspects of the materials, allowing for the capturing of a multidimensional representation of how visual material discrimination is processed. During the Verbal Description task, the same group of participants provided unrestricted descriptions of the material with texts covering five aspects: material name, color, optical properties, mechanical properties, and surface texture. These aspects have been found useful in characterizing the mental representations of materials [44].

We constructed the Representational Dissimilarity Matrices (RDMs) from each participant's behavioral results for both tasks. A Vision RDM is created based on the on-screen Euclidean distances of pairwise comparisons of material similarity [49] from the Multiple Arrangement. Meanwhile, we also built a Text RDM by encoding the images' text descriptions provided by the participant into an embedding space with a pre-trained LLM (see Methods). We tested four publicly accessible LLMs, CLIP's text encoder [7], Sentence-BERT [51], GPT-2 [52], and OpenAI Embedding V3-small, whose embedding spaces were shown to capture the semantic similarity of textual information. The primary analysis in this paper is conducted using the CLIP's text embedding unless otherwise noted.

Across individuals, we found a moderate correlation between the RDMs of the two tasks within each participant, by applying the Representational Similarity Analysis (RSA). Fig 4A displays the RDMs of three participants (see S2 Fig for all participants' results). While the participants used different numbers of unique words (Fig 5A, mean = 128 unique words, max = 288 words, min = 37 words), we found that all of the participants' verbal responses exhibited a significant correlation (min Spearman's correlation $r_s$ = 0.09, max Spearman's correlation $r_s$ = 0.53, all $p < 0.001$, FDR-corrected) with their own multiple arrangement behavior, signifying the presence of inherent cross-task consistency within an individual (Fig 5B "With material name" condition). These moderate correlations reflect that participants' own Vision and Text RDMs share similarities in their overall structures, while also underpinning differences in their local patterns. We observed a stronger correlation when comparing the group average Vision and Text RDMs (Spearman's correlation $r_s$ = 0.74, $p < 0.001$) (rightmost column in Fig 4A). By applying classical multidimensional scaling (MDS) on the group average RDMs, we found that Vision and Text embeddings exhibit similar organizations, forming three major clusters: squishy-like (squishy toys, top left cluster in MDS), soap-like (soap and soap-to-toy, bottom left cluster in MDS), and rock-like (rock, rock-to-toy, and soap-to-rock, bottom right cluster in MDS).

## Vision- and language-based representations reveal salient semantic features

Next, we sought to interpret the representative dimensions of materials expressed through the behavioral tasks. We annotated the MDS results of the group average Vision RDMs with the image stimuli and the participants' verbal descriptions. Colorfulness, material name, and softness are the key features across participants. The "Colorfulness" panel in Fig 6 shows that materials with vivid body colors and high saturation (left side in MDS, e.g., red, pink) are separated from less saturated colors (e.g., light blue and light gray). Materials' chemical and physical properties determine the specific range of their colors and surface textures [34]. This innate connection may facilitate visual material categorization. To further comprehend the material clustering in the MDS, we labeled the most frequent word participants used in verbal descriptions for each stimulus. In the description of "material name" and "mechanical properties", we observed that participants tend to group "hard" materials (e.g., rock, glass, or crystal) away from "soft" ones (e.g., squishy toy, or rubber) ("Mechanical properties" and "Material name"

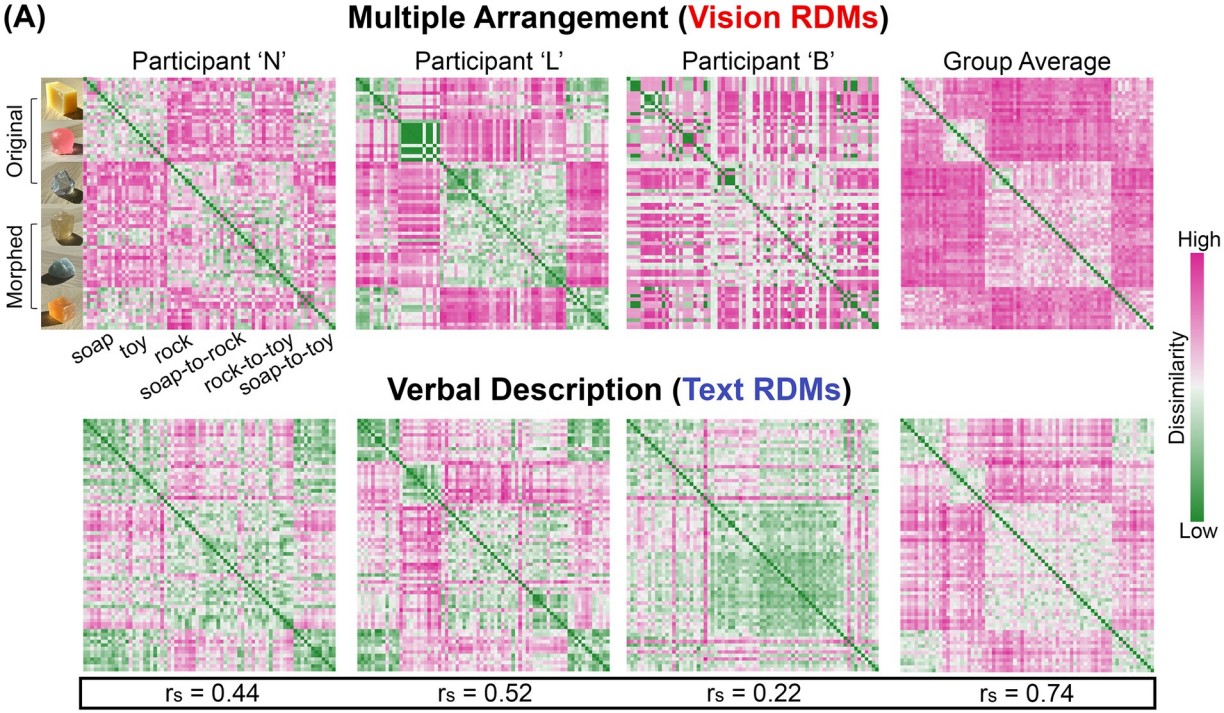

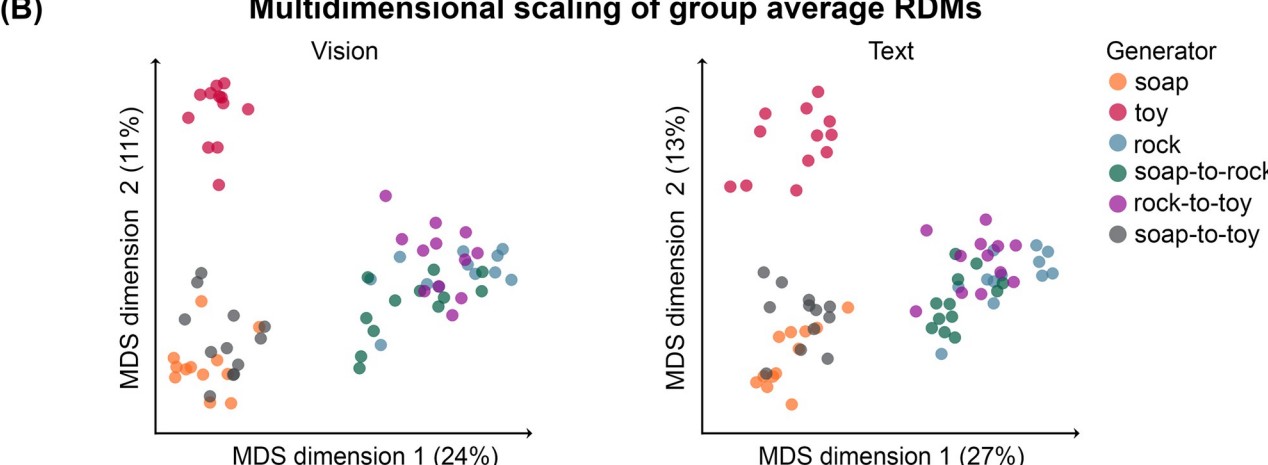

**Fig 4. Vision-based similarity judgment and verbal description of materials are moderately correlated.** (A) RDMs of visual material similarity judgment via Multiple Arrangement (Vision RDMs) and Verbal Description (Text RDMs). Top: Vision RDMs. Bottom: Text RDMs. From left to right: RDMs for three participants and the group average RDM across all participants. In each RDM, on both x- and y-axis, the images are organized by the type of material generator, spanning from the learned original materials (i.e., soap, toy, rock) to the morphed midpoint materials (i.e., soap-to-rock, rock-to-toy, and soap-to-toy). The dissimilarities in the individual RDMs are normalized to the 0 and 1 range, and the average RDMs are computed based on the normalized values. The green colors indicate low dissimilarity between pairwise combinations of materials, whereas the pink colors indicate high dissimilarity. The Spearman's correlation ($r_s$) between the corresponding Vision and Text RDMs are annotated in the box below. (B) Two-dimensional embedding from the MDS of the group average Vision and Text RDMs, color-coded based on the six types of material generator depicted in (A). The percentage of explained variance is shown in parentheses.

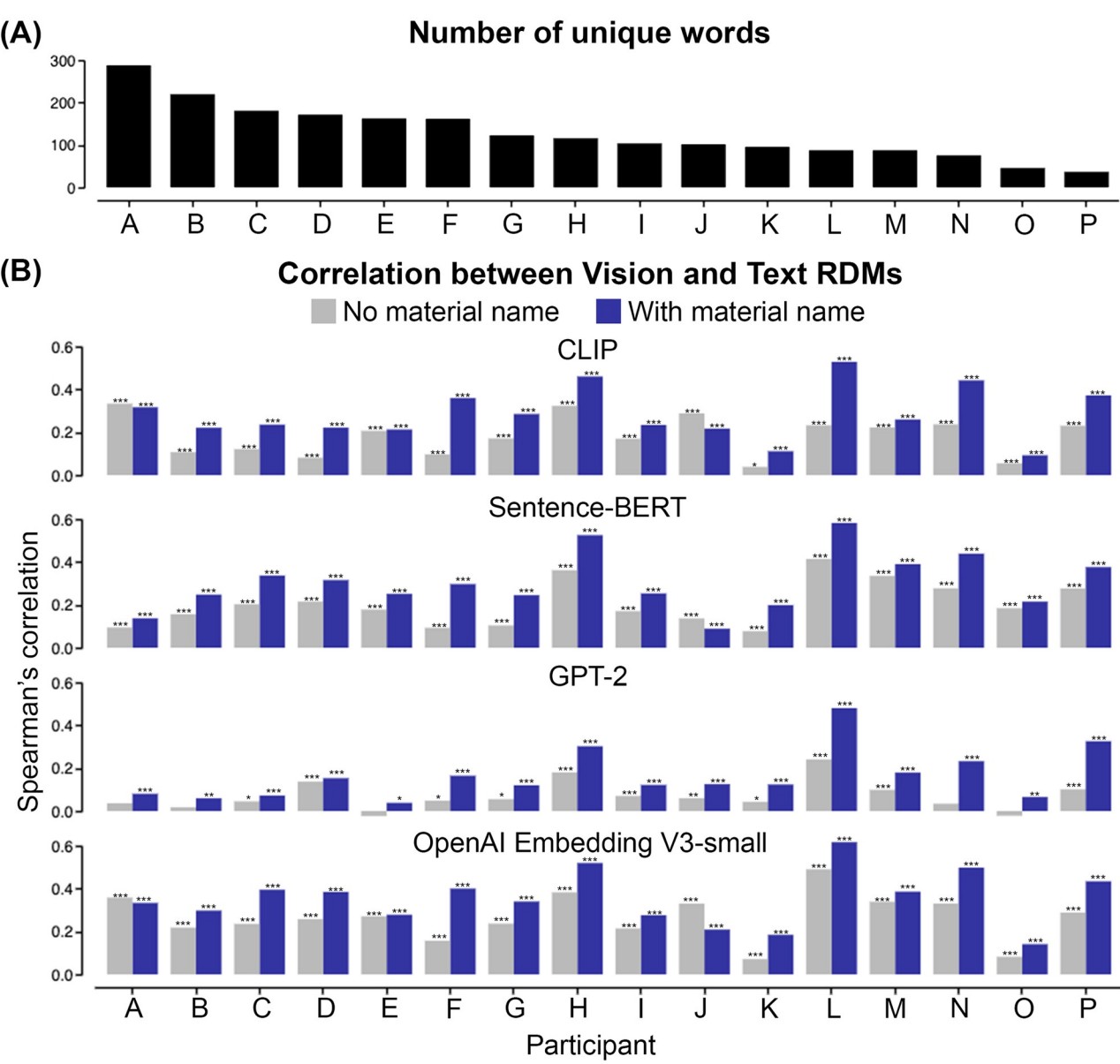

**Fig 5. Effect of the number of unique words, language models, and material names on individual behavioral results.** (A) Distribution of the number of unique words participants used in the Verbal Description task. (B) Comparison of vision-language correlations across different language models. For each individual, we computed within-person Spearman's correlation between the Vision and Text RDMs. The Text RDM is built by embedding verbal descriptions with four different pre-trained LLMs: CLIP's text encoder, Sentence-BERT, GPT-2, and OpenAI Embedding V3-small. The blue bars indicate the correlation values when all text features are included to construct the Text RDM. The gray bars indicate the correlation values when the "material name" is excluded from constructing the Text RDM. Asterisks indicate FDR-corrected p-values: *** $p < 0.001$, ** $p < 0.01$, and * $p < 0.05$.

panels in Fig 6). Here, perceived softness might be a notable attribute associated with the material category. On the other hand, the descriptive words representing visual characteristics, "optical properties" and "surface texture" (e.g., "translucent", "opaque", "rough", and "crack"), are dispersed across different major clusters ("Optical properties" and "Surface texture" panels in Fig 6). Semantic features of the material may have various levels of functional importance, depending on the nature of the attribute. While colorfulness, material name, and softness

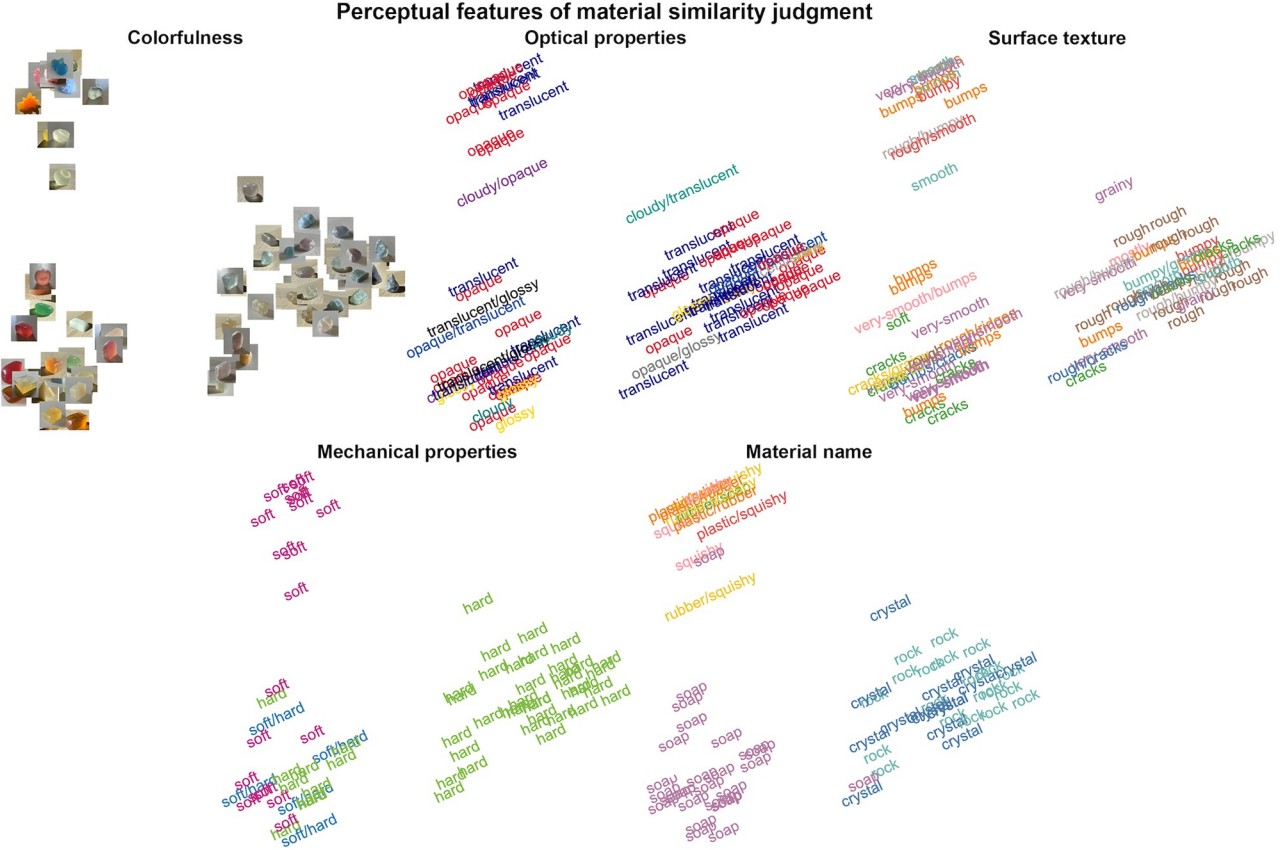

**Fig 6. Annotated MDS of the group average Vision RDM.** For "Colorfulness", we display the data point in the form of the image stimuli. For "Optical properties", "Surface texture", "Mechanical properties" and "Material name", we display the most frequently used word (applied with the same color) for each image, aggregated across all participants. The color schemes of words are not comparable across MDS plots. An interactive version of this plot is provided in the S1 File.

establish the coarse-grained material assessment, optical and surface properties could further support fine-grained material discrimination.

We further found that removing the "material names" from the text embeddings ("No material name" condition in Fig 5B) significantly decreased the correlation between Vision and Text RDMs for almost all participants (Wilcoxon one-sided signed-rank test, all $p < 0.0005$ across four tested language models). Different LLMs produced similar results, except GPT-2 embedding led to lower vision-language correlations. We observed a similar effect when separately comparing the perceptual RDMs of the 36 images of "original" and 36 images of "morphed" materials. Removing "Material Name" also reduces vision-language correlation in "morphed" materials (see S4 Fig). Material naming may serve as a high-level feature that envelops the particular structural combination of various material characteristics, providing critical information to the perceptual inference of material attributes.

## Comparing stimulus-level similarity structures between vision and text

To further examine the relationship between representation from vision and language at the stimulus level, we use an unsupervised alignment method, Gromov-Wasserstein Optimal

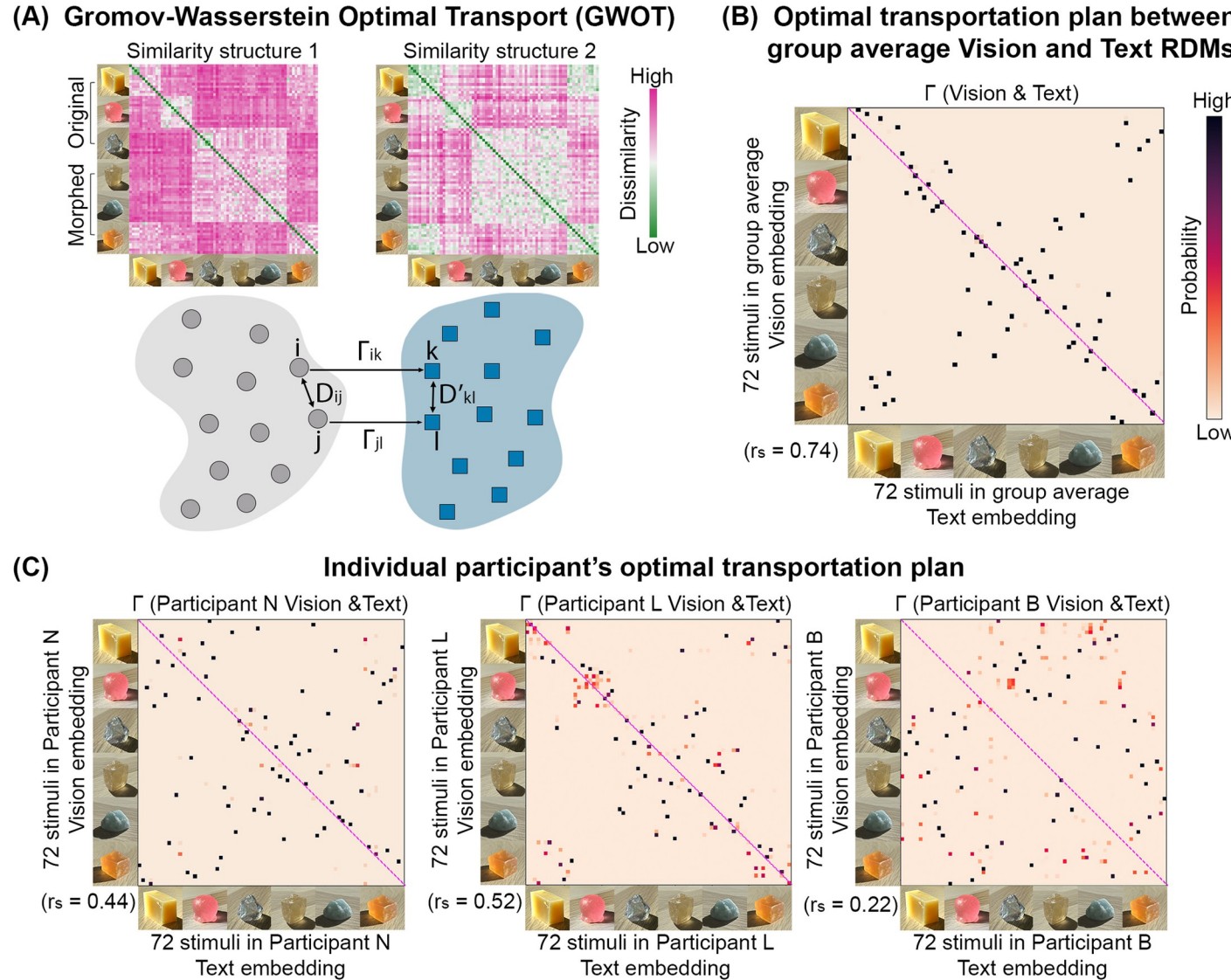

**Fig 7. The similarity structures between visual judgment and verbal description align on the coarse categorical level but lack precise one-to-one stimulus-level mapping.** (A) Illustration of Gromov-Wasserstein Optimal Transport (GWOT), an unsupervised alignment method to compare two similarity structures (e.g., similarity structure 1 and 2). $D_{ij}$ denotes the dissimilarity between stimulus $i$ and $j$ in one RDM, and $D'_{kl}$ denotes the dissimilarity between stimulus $k$ and $l$ in another RDM. Solving the minimization problem of Gromov-Wasserstein distance (GWD) yields an optimal transportation plan $\Gamma$. The group average Vision (Top left) and Text (Top right) RDMs are shown as examples. (B) Optimal transportation plan $\Gamma$ between group average Vision RDM and Text RDM. Each element in the $\Gamma$ matrix indicates the probability of an image in the similarity structure of verbal description corresponding to another item in the similarity structure of visual material similarity judgment. The purple diagonal indicates the perfect alignment on the image-to-image level. Only a small fraction of diagonal elements in $\Gamma$ show high values, indicating the lack of one-to-one mapping between verbal descriptions and visual judgment of the stimuli. The Spearman's correlation ($r_s$) between the Vision and Text RDMs is noted in the bottom left corner of the $\Gamma$ matrix. (C) $\Gamma$ matrix computed from individual participant's Vision and Text RDMs. The samples on the X- and Y-axis follow the same order as the Vision and Text RDMs.

Transport (GWOT), to scrutinize structural alignment between the perceptual spaces from visual judgment and verbal description [53, 54] (see Fig 7A and Method). Given two embeddings, GWOT provides a mapping between them based only on their internal relationships (i.e., distances between data points). It seeks to identify an optimal transportation plan $\Gamma$ between these two similarity structures (i.e., expressed through the dissimilarity matrices), by

solving the optimization problem on Gromov-Wasserstein distance (GWD):

$$GWD = \min_{\Gamma} \sum_{i,j,k,l} ||D_{ij} - D'_{kl}||^2 \Gamma_{ik} \Gamma_{jl} \tag{1}$$

Applying GWOT to the group average Vision and Text RDMs, we discovered that vision—and language-based representational spaces are aligned at the coarse categorical level but lack precise one-to-one mapping on the stimulus level. Fig 7B shows the optimal transportation plan matrix Γ (between group average Vision and Text RDMs). Each element in Γ indicates the probability of a sample in one similarity structure corresponding to another in the other similarity structure. We observed that Γ significantly deviates from being a diagonal matrix. Significant deviations typically occur within materials of similar visual appearance, such as samples in the same coarse material categories (e.g., within soaps). The misalignments are enlarged among the morphed material, especially those morphed from rocks (e.g., soap-to-rock, rock-to-toy). We observed a marginal enhancement in stimulus-level vision-language alignment when employing a semantically richer text embedding, such as OpenAI Embedding V3-small model, yet misalignment in local structures still persists (S3 Fig).

Fig 7C shows significant differences in the stimulus-level structures among the perceptual RDMs from different participants. The figure shows that even for participants exhibiting a substantial correlation between Vision and Text RDMs (e.g., Participant L), their Γ matrix has considerable data points that deviate from the diagonal line (Fig 7C middle panel). We computed the top-1 matching rate to quantify the stimulus-level alignment across individuals. For each stimulus, we consider it as a match if the transportation plan assigns the highest probability between the same stimuli in the two similarity structures. Contrary to the top-1 matching rate for the group average Vision and Text RDMs (12.5%), the individual participants' RDMs tend to have a lower top-1 matching rate (ranging from 1.38% and 6.94%), suggesting the significant structure difference between vision and language representations at the individual level.

We assessed the inter-participant consistency of the behavioral tasks with the leave-one-out test by iteratively correlating one participant's data with the group average of the rest of the participants. We found significant (all $p < 0.001$) inter-participant correlations in both tasks, with Vision RDMs (mean correlation $r = 0.41$) displaying a greater variance than Text RDMs (mean correlation $r = 0.57$). Compared with the ways of articulating materials with words, participants tended to be more diverse in visually judging material similarities. Such variation in visual judgment strategies may contribute to the enlarged vision-language misalignment on the individual-participant level.

## Image representations from the pre-trained data-rich models capture material features

Our findings reveal subtle yet discernible visual differences among material samples that are challenging to articulate verbally. Next, we examined whether the image representations of materials extracted from pre-trained data-rich models correlate with human visual perception and explored the plausibility of using the distilled image features from such models to narrow the gap.

We examined the image representation derived from the pre-trained models: self-supervised vision model (e.g., DINO), and visual-semantic model (e.g., CLIP and OpenCLIP [50]). Fig 8A Middle and Right panels show the constructed Image-feature RDMs by encoding our psychophysical stimuli with the tested models (see detail in Method). We then computed the

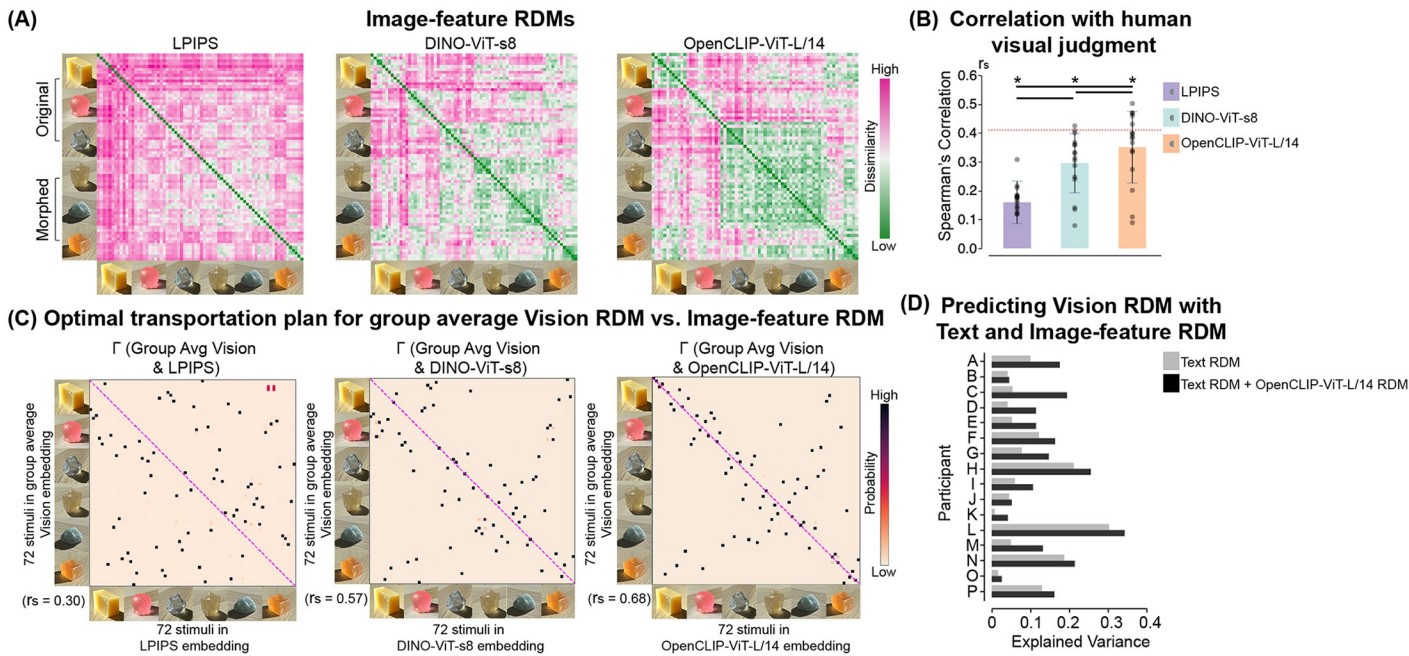

**Fig 8. Compare human visual judgments with image representations.** (A) Image-feature RDMs of 72 stimuli created from pre-trained models: LPIPS (perceptual image similarity), DINO-ViT-s8 (self-supervised model), OpenCLIP-ViT-L/14 (visual-semantic model). The elements in the RDM follow the same order as the Vision and Text RDMs. (B) Spearman's correlation between an individual's Vision RDM and Image-feature RDM from each vision encoder. The bars represent the average correlations across participants. The block dots represent the individual participants. The red dotted line indicates the lower bounds of the noise ceiling of human visual judgment results. On top of each bar, * indicates $p < 0.005$ for model-specific one-sided signed-rank tests against zero. The horizontal black bar indicates $p < 0.05$ for two-sided pairwise signed-rank tests between vision encoder models. (C) Optimal transportation plan ($\Gamma$) matrix between group average human visual judgment and image feature embedding of 72 psychophysical stimuli. The Spearman's correlation (rs) between the Image-feature RDM and the group average human Vision RDM is marked in the bottom left corner. (D) Jointing verbal descriptions with image features from the visual-semantic model OpenCLIP-ViT-L/14 improves the prediction of visual material similarity judgment for all participants. The y-axis represents individual participants. The x-axis represents the explained variance.

Spearman's correlations between each Image-feature RDM and each participant's Vision RDM. Fig 8B shows that the deep image-feature representations from the data-rich model models moderately correlated with human visual judgment (DINO-ViT-s8: mean $r_s = 0.29$, W(16) = 136, $p < 0.0001$; OpenCLIP-ViT-L/14: mean $r_s = 0.35$, W(16) = 136, $p < 0.0001$). Compared with the pure vision model DINO, the visual-semantic model OpenCLIP-ViT-L/14, which was trained on a larger dataset (over 2 billion English image-text pairs), had a higher correlation (W(16) = 2, $p = 0.00009$) with human visual judgments. In contrast, the relatively low-level image feature similarity, such as the features from the perceptual similarity metric LPIPS (mean $r_s = 0.16$, W(16) = 135, $p < 0.0001$), is less correlated with human behavior results. Our results indicate the general visual representation captured by the large-scale contrastive vision-language pretraining is transferable to material-related visual tasks. Comparing image representation from other weakly-supervised [7], self-supervised models [55] and perception-based image similarity metrics [56, 57], we discovered the semantically-aware model (e.g., OpenCLIP) could best predict human visual similarity judgment (see S6 Fig).

We illustrated the alignment and misalignment between humans and networks by applying GWOT on the group average Vision RDM and each model-based Image-feature RDM. The resulting $\Gamma$ matrix from the semantically-aware model, OpenCLIP-ViT-L/14 (Fig 8C Rightmost), shows that the model's representation roughly aligns with human perception at the level of coarse categories ($r_s = 0.68$, $p < 0.005$), yet precise mapping at the stimulus-level is still lacking (top-1 matching rate 15.27%). As shown in Fig 8C Leftmost and Middle panels, low-level image feature and DINO's deep image feature representation have larger structural

deviance from human visual judgment than those of the OpenCLIP-ViT-L/14 model. This demonstrates that the weakly-supervised model, which learns high-level semantic correlations between images and text, may capture perceptually relevant visual features crucial to material perception.

Lastly, we tested whether combining the model-extracted image representations and human verbal descriptions improves the prediction of human visual judgments of materials. For each individual, we used the Image-feature RDM (based on OpenCLIP ViT-L/14, Fig 8A Rightmost panel) and the participant's Text RDM together (i.e., full model) or only the Text RDM (i.e., reduced model) as predictors in a multiple regression model to predict the participant's Vision RDM. Compared with the reduced model, the model incorporating verbal descriptions and image features (i.e., Text RDM + OpenCLIP ViT-L/14 RDM) performs significantly better ($p < 0.0001$ with ANOVA test) and increases the explained variance (adjusted $R^2$) across all participants (Fig 8D). This suggests combining human text with image features from large-scale, pre-trained, semantic-aware models might narrow the gap between visual and semantic representations in material perception.

## Discussion

We propose a framework to probe the link between vision and language using AI-generated images of familiar and ambiguous materials. By combining multimodal psychophysics with machine learning techniques, our approach offers a quantitative exploration of the representational space of materials. Specifically, using an unsupervised image synthesis model, we developed an efficient approach to create a diverse array of plausible visual appearances of familiar and unfamiliar materials. With these images, we measured and analyzed behavioral tasks alongside image features derived from pre-trained data-rich models. We found a moderate but significant correlation between visual judgments and verbal descriptions of materials within individual participants, signifying both the efficacy and limitation of language in describing materials.

The lack of precise alignment between the representations from two behavioral tasks pinpoints the gap between visual judgment and verbal description of materials (Fig 7C). On the one hand, combining human vision- and text-based representations reveals informative features for material discrimination, such as the object's color, softness, and material name. This would be challenging to manifest when limited to a single modality. At the same time, the verbal descriptions do not fully capture the visual nuances of material appearances, which are more pronounced between materials samples within a category and between ambiguous materials. One potential explanation could be that participants faced difficulty in describing subtle visual characteristics, such as spatial color variation and surface geometric complexity, with accuracy and consistency. Nevertheless, these visual attributes could be crucial in finely distinguishing samples within the general clustering of materials. These results highlight that visual representations of materials are richer than verbalized semantic features, underscoring the differential roles of language and vision in perception. This notion is crucial in developing computer vision applications, from material-related scene annotation to text-guided image synthesis.

Our results show that when the material name is removed from the text embedding, the correlation between Vision and Text RDMs systematically decreases across participants (Fig 5). We conducted the MDS and GWOT analysis with the human text embedding at the "No Material Name" condition (see S5 Fig). We observed that removing the material name may further reduce the one-to-one mapping between verbal description and visual judgment of material (Spearman's correlation $r_s = 0.74$ for the "With Material Name" condition to $r_s = 0.50$ for the "No Material Name" condition). This may stem from the functional roles that

nouns (material names) play in everyday language usage [58] and highlights the significant role of material names in visual categorization. With material names (e.g., crystal or soap), we can label materials that possess an array of unique and/or related attributes, such as softness, translucency, glossiness, and the object's shape. During this labeling process, material names can encapsulate the perceptual similarity of materials across multiple dimensions and partition samples of materials into a system of semantic categories [59], potentially serving as an efficient approach to communicating about material appearance. In future works, we might investigate material perception from the perspective of effective communication [4, 59, 60], such as examining the structure and complexity of material naming and comparing material naming across various language systems.

Our investigation of the image representation from the data-rich models shows the usefulness of the learned task-agnostic features in visual material reasoning. Through pre-training on large-scale datasets, we found most of our tested models can approximate an "average" human participant's visual material similarity structure without being explicitly trained on material-specific tasks. This shows that the generalized visual features can cluster materials into coarse categories similar to humans. Furthermore, our findings suggest the weakly-supervised vision-semantic models (e.g., CLIP, OpenCLIP), which learn high-level semantic correlations between images and texts, are more likely to capture perceptually relevant visual features crucial to material perception. These computational models may provide insights into searching vision-specific nuanced features in material perception, potentially narrowing the gap between vision and language. Recent studies have demonstrated the possibility of improving the representation of visual-semantic models using human perceptual judgment as supervision [61]. Future investigations could explore whether this approach is applicable to uncovering critical information related to human visual judgments of materials (e.g., material property estimation).

Evaluating the link between visual judgment and verbal description with behavioral data is the first step for probing their neural representations in material perception. Neuroscience research actively examined the neural representation of language and non-linguistic processing (e.g., music, working memory) and investigated the specificity and interrelationship of brain regions responsible for these cognitive skills [62]. Efforts were also made to explore how the brain encodes certain conceptual representations (e.g., objects, actions) elicited by visual and linguistic stimuli [15, 63]. Recent work suggested that incorporating language feedback is crucial for explaining neural responses in high-level visual brain regions [64]. Following these practices, a plausible future direction could be to examine whether and how brain regions' engagement for visual judgment differs from those activated by semantic descriptions of materials. Addressing such cortical representations across modalities may help to unravel the open questions: What is the causal relationship between material recognition and attribute estimation? At a more fundamental level, how does the functional mechanism of perceiving materials differ from and connect to that of perceiving textures and objects?

Unlike previous works that usually examined material perception with real or rendered zoom-in surfaces, we intentionally synthesized images of materials coupling with object-level realism. Our approach of constructing a Space of Morphable Material Appearance with transfer learning and model interpolation methods could be extended to a broader range of materials (e.g., metal, glass). Beyond sampling at the interpolation midpoint, the expressiveness of our model and its latent representation offers a unique capability to manipulate material-related attributes (e.g., translucency and surface geometry) of the object. This facilitates controlled and continuous adjustments of visual characteristics linked to material categories. This capability enables us to investigate the relationship between visual features and categorical perception, which is crucial in understanding the interplay between perception and language. This intersection has been actively studied in color perception [65].

In our current Verbal Description experiment, we provided a predetermined list of material attributes to encourage the participants to actively consider the material attributes. Given that our stimuli only contain a single object in a simple scene, the design is suitable for collecting verbal communication of materials. In the future, we will consider more free-form, even complete free-form expression when we create stimuli of more complex scenes. Our current dataset focuses on translucent materials, which share certain visual characteristics. We acknowledge that the current dataset of our selection of material could be restricted in its diversity. Our effort is a step towards building a more comprehensive large-scale real photo dataset of diverse materials.

In conclusion, by demonstrating the potential link between vision- and language-based representations, we reveal salient features related to material perception, while also highlighting the disparity between semantic and visual representations. Our study invites further investigation of material perception by considering it an avenue to explore the language-perception relationship across a broad range of visual cognition tasks.

## Methods

**Ethics statement**. The experiments were approved by the ethics board at the American University (AU) (Protocol number IRB-2020–155) and conducted in adherence to the Declaration of Helsinki.

### Image datasets

We created our training datasets of high-resolution images (1024 pixels × 1024 pixels) by taking photographs of real-world materials with an iPhone 12 Mini smartphone. Overall, our training data consists of three subcategories: soap ($D_{soap}$), rock ($D_{rock}$), and squishy toy ($D_{toy}$) datasets, including 8085 (60 objects), 3180 (24 objects), and 1900 (15 objects) images, respectively.

### StyleGAN and transfer learning

We used the style-based generative adversarial network, StyleGAN2-ADA, as the backbone model. Our previous work, Liao et al. (2023) [29], provides a detailed description of the model and the training process. StyleGAN2-ADA inherently applies a variety of data augmentation during training, and the length of training is defined by the total number of real images seen by the network. We obtained a Soap Model by training the StyleGAN2-ADA from scratch on $D_{soap}$ for a total length of 3,836,000 images, with a learning rate of 0.002 and $R_1$ regularization of 10.

We fine-tuned the Soap Model separately on the $D_{rock}$ and $D_{toy}$, which allows all model parameters to adjust to the new datasets. Full-model fine-tuning processes on $D_{rock}$ and $D_{toy}$ used the same hyperparameters as the training on $D_{soap}$. The lengths of fine-tuning were 1,060,000 and 960,000 images for $D_{rock}$ and $D_{toy}$, respectively. We used the models with the lowest Fréchet Inception Distance (FID) scores for the rest of our study. The FID scores for Rock and Toy Models are 22.22 and 23.38, respectively. All training was performed on a Tesla V100 GPU on Google Colab.

### Cross-category material morphing

The morphing of images of materials requires applying linear interpolation of the layer-wise latent codes $w \in W$, as well as the StyleGAN's generator weights [66]. To morph from a source to a target material, we first sample two latent codes (i.e., $w_{source}$ and $w_{target}$) from the corresponding learned $W$ latent spaces (e.g., $w_{soap} \in W_{soap}$ as source, $w_{rock} \in W_{rock}$ as target). As

illustrated in Fig 2C, $w$ is a tensor with the dimension of $18 \times 512$. With Eq 2, we can compute the interpolated latent code $w_\lambda$, at any desired step size $\lambda$. The dimension of $w_\lambda$ is also $18 \times 512$. Similarly, we implement linear interpolation between the convolutional weights of each convolution layer in the source material generator and the corresponding weights in the target material generator. The weights are multidimensional tensors. With the same $\lambda$, we calculate the interpolated generator weights $G_\lambda$ (Eq 3).

$$w_\lambda = w_{source} + \lambda(w_{target} - w_{source}) \tag{2}$$

$$G_\lambda = G_{source} + \lambda(G_{target} - G_{source}) \tag{3}$$

We insert $w_\lambda$ into $G_\lambda$ to generate the image of morphed material. Specifically, each of 18 slices of $w_\lambda$ is injected into the convolution layer at the corresponding spatial resolution (from 4 pixels $\times$ 4 pixels to 1024 pixels $\times$ 1024 pixels) (Fig 2C).

## Psychophysical experiments

**Participants.** Sixteen participants (13 female, median age = 22) from AU participated in the experiments with written consent and they were reimbursed for their participation. All were Native English speakers and had a normal or corrected-to-normal vision.

**Stimulus selection.** We first generated 30 images for each of the "original" materials: soaps, rocks, and squishy toys, by sampling from their corresponding latent spaces, $W_{soap}$, $W_{rock}$, and $W_{toy}$ and synthesizing with their paired material generators, $G_{soap}$, $G_{rock}$, and $G_{toy}$. We balanced the images in two lighting conditions for each material category: strong and weak (Fig 2E).

We randomly paired up two different "original" materials under the same lighting conditions and then synthesized the image corresponding to the linear interpolation midpoint (step $\lambda = 0.5$). We initially generated 1000 images of morphed materials through the corresponding midpoint material generators ($G_{soap-to-rock}$, $G_{soap-to-toy}$, and $G_{rock-to-toy}$). We picked 12 images synthesized from six material categories: soap, rock, squishy toy, soap-to-rock midpoint, rock-to-toy midpoint, and soap-to-toy midpoint. For each material category, half of the selected images are from strong lighting conditions (i.e., sunny indoor scene), and the remaining half are from weak lighting conditions (i.e., overcast indoor scene). We selected 72 images and tried to make the range of visual appearances as diverse and natural as possible. These images were then used as stimuli for Multiple Arrangement and Verbal Description tasks.

**Multiple Arrangement task.** We conducted the multiple arrangement experiment using Meadows.com (https://meadows-research.com/). Participants were instructed to arrange the images (180 pixels $\times$ 180 pixels) of materials based on the "similarity of material properties" by dragging and dropping them in the circled region (Fig 3A). In the first trial, the participants roughly arranged all 72 images into groups. In the subsequent trials, more refined subsets were chosen and displayed by an adaptive lift-the-weakest algorithm to reduce the remaining uncertainty of the similarity judgment of materials [49]. The average duration of the experiment was about 60 minutes. The pairwise on-screen Euclidean distances between the arranged images were computed upon the completion of the experiment, producing a Vision RDM with inverse MDS.

**Verbal Description task.** With the same 72 images used in the Multiple Arrangement task, participants described the material in the image by freely inputting texts based on five aspects (see Fig 3B). The stimuli were presented in size of 512 pixels $\times$ 512 pixels. They had unlimited time on each trial and were not restricted regarding the order in which they could enter their responses. In the experiment instruction, the participants were told, "Please provide

keywords or short sentences to describe the material from the following aspects: The name of the material, color, optical properties, mechanical properties, and surface texture." We also encouraged the participants to describe the material attributes in the way they wanted and assured them that the task was subjective, without correct answers.

**Experiment procedures.**   All participants first completed the Multiple Arrangement task, and then the Verbal Description task in a separate session. All experiments were conducted in a dimly lit laboratory room. The stimuli were presented on an Apple iMac computer with a 21.5-inch Retina Display, with a resolution of 1920 pixels × 1080 pixels.

**Creating Text RDMs from verbal description data.**   We used a fixed template to concatenate the five aspects that participants described an image: "It is a material of [material name] with the color of [color], it is [optical properties], it is [mechanical properties], and it is [surface texture]." Next, we encoded the concatenated text into a feature vector through a pre-trained LLM. The four commonly used pre-trained transformer-based language models (i.e., CLIP's text encoder [7], Sentence-BERT [51], GPT-2 [52], and OpenAI Embedding V3-small), can embed a sentence or paragraph of text into a high-dimensional feature space. For each model, we extracted the feature vector at the last hidden layer. The size of the embedded text feature vector varies across different language models: 512 for CLIP, 384 for Sentence-BERT, 768 for GPT-2, and 1536 for OpenAI Embedding V3-small. For each participant, we built a $72 \times 72$ Text RDM by computing the pairwise Cosine distance between the resulting feature vectors of the verbal descriptions (Fig 4A, Bottom Row).

To investigate the effect of removing the "material name" on the embedding of the verbal descriptions, we used the following template to form the image caption: "It is a material with the color of [color], it is [optical properties], it is [mechanical properties], and it is [surface texture]." Hence, we used the same procedure described above to encode the descriptions without material names as feature vectors.

**Gromov-Wasserstein Optimal Transport (GWOT).**   GWOT is an unsupervised alignment technique, identifying the best transport strategy between point clouds across two domains, while the correspondence between data in two similarity structures is not assumed.

As illustrated in Fig 7A, given two similarity structures, represented by their RDMs (normalized to 0 to 1 range), $D$ and $D'$, GWOT seeks to optimize the Gromov-Wasserstein distance (GWD). To enhance the optimization efficiency of GWD, an entropy-regularization term $H$ ($\Gamma$) is added to Eq (1). This regularized GWD, $GWD_\epsilon$, is optimized as the following:

$$GWD_\epsilon = \min_\Gamma \sum_{i,j,k,l} ||D_{ij} - D'_{kl}||^2 \Gamma_{ik} \Gamma_{jl} + \epsilon H(\Gamma), \tag{4}$$

where $\Gamma$ is the optimal transportation plan.

A local minima can be found by solving the above optimization problem. We performed hyperparameter tuning on $\epsilon$, searching for its optimal value over 0.0001 to 0.1 with logarithmic spacing (500 different $\epsilon$ values). For each searched $\epsilon$, we can find its corresponding $\Gamma$. We finally selected the $\Gamma$ that minimized GWD (without the entropy-regularization term (Eq 1)) as the optimal transportation plan. A detailed explanation of GWOT can be found in [53, 54].

**Creating Image-feature RDM with pre-trained models.**   We extracted the latent image features of each stimulus in the psychophysical experiments with pre-trained data-rich models. The weakly-supervised models, OpenCLIP, were pre-trained on a large-scale image-text-pair dataset, LAION-2B. They jointly trained an image encoder and a text encoder to maximize the cosine similarity of the image and text embeddings in order to predict which images were paired with which texts in the training dataset. With the Vision Transformer (ViT) based image encoder of OpenCLIP (OpenCLIP-ViT-L/14), we extracted the embedding of our

stimuli based on the feature vector at the final linear projection layer. In contrast, DINO, the self-supervised training of ViT, is trained on images only. It uses a knowledge-distillation learning paradigm and was pre-trained on the ImageNet dataset. We extracted the feature vector from the output layer of the pre-trained DINO-ViT-s8. The size of the image feature vectors varies depending on the pre-trained models (see S1 Table). We computed the pairwise Cosine distances between images based on the model-extracted features to build the Image-feature RDMs (see examples in Fig 8A).

**Predicting Vision RDM with Text and Image-feature RDMs.** To determine the contribution of verbal description and image-level features to the multiple arrangement behavior, we use multiple linear regression based on the Text and Image-feature RDMs derived from various image encoders [67].

We first converted the $72 \times 72$ RDMs into 2556-dimensional feature vectors, by extracting the off-diagonal elements of an RDM. We set the participant's own Vision RDM feature vector as the predicted variable. Two first-order multiple regression models were fitted for each participant: one "full" model that included the Text RDM and the Image-feature RDM; and one "reduced" model that only included the Text RDM. For each model, we computed the adjusted $R^2$ to indicate the explained variance of the Vision RDM. We also used ANOVA to test whether the "full" model improves the fit of the data compared to the "reduced" model, with statistical significance (95% confidence level).

## Supporting information

**S1 Text. Explanatory texts for each figure and table in Supporting information.**
(PDF)

**S1 Fig. Seventy-two generated images used as stimuli for both Multiple Arrangement and Verbal Description experiments.**
(TIF)

**S2 Fig. Individual participant's (N = 16) RDMs of visual material similarity judgment via Multiple Arrangement (Vision RDM) and Verbal Description (Text RDM).** The Text RDMs are based on the CLIP's text embedding results, as illustrated in the main paper Fig 4A. The Spearman's correlation ($r_s$) between the participant's own Vision and Text RDMs is marked on top of each pair of RDMs.
(TIF)

**S3 Fig. Optimal transportation plan between Vision and Text RDMs. The Text RDMs are based on OpenAI Embedding V3-small.** (A) Optimal transportation plan matrix ($\Gamma$) between group average Vision and Text RDMs. (B) Optimal transportation plan matrix of individual participant's Vision and Text RDMs. The Spearman's correlation ($r_s$) between the Vision and Text RDMs is noted in the bottom left corner of the $\Gamma$ matrix.
(TIF)

**S4 Fig. Correlation between Vision and Text RDMs with ROI for the original "original" (soap, rock, toy) and "morphed" (soap-to-rock, soap-to-toy, toy-to-rock) materials.** Top: text embedding derived from CLIP's text encoder. Bottom: text embedding derived from OpenAI Embedding V3-small. The blue bars indicate Spearman's correlation values when all text features are included to construct the Text RDM. The gray bars indicate the correlation values when the "material name" is excluded from constructing the Text RDM. Asterisks indicate FDR-corrected p-values: *** $p < 0.001$, ** $p < 0.01$, and * $p < 0.05$.
(TIF)

**S5 Fig. Representational space of human text response in the "No Material Name" versus "With Material Name" condition based on CLIP's text embedding.** Under each text embedding condition, the leftmost column shows the group average Text RDMs. The middle column is the MDS embeddings of the group average Text RDMs. The rightmost column shows the optimal transportation plans that compare the group average human Vision RDM (from the Multiple Arrangement task) with the group average Text RDM. (A) "No Material Name" (when the "material name" is removed from the text embedding) (B) "With Material Name" condition.
(TIF)

**S6 Fig. Spearman's correlation between an individual's Vision RDM and Image-feature RDM from each tested vision encoder.** The bars represent the average correlations across participants. The block dots represent the individual participants. The red dotted line indicates the lower bounds of the noise ceiling of human visual judgment results. On top of each bar, * indicates $p < 0.005$ for model-specific one-sided signed-rank tests against zero. The horizontal black bar indicates $p < 0.05$ for two-sided pairwise signed-rank tests between two nearby vision encoder models shown in the plot.
(TIF)

**S7 Fig. Cross-material morphing examples.** The source material transforms into the target material with a nine-step interpolation.
(TIF)

**S1 Table. Size of the latent feature vectors from the pre-trained vision and LLM models.**
(TIF)

**S1 File. Interactive plots of Fig 6 of our main paper. Annotated MDS of the group average Vision RDM**. For each stimulus, we annotated with the aggregated human text description along with the word frequency of each material aspect, "colorfulness", "optical properties", "surface texture", "mechanical properties", and "material name."
(ZIP)

## Author Contributions

**Conceptualization:** Chenxi Liao, Masataka Sawayama, Bei Xiao.

**Data curation:** Chenxi Liao.

**Formal analysis:** Chenxi Liao.

**Funding acquisition:** Bei Xiao.

**Investigation:** Chenxi Liao, Bei Xiao.

**Methodology:** Chenxi Liao, Masataka Sawayama, Bei Xiao.

**Project administration:** Bei Xiao.

**Resources:** Bei Xiao.

**Software:** Chenxi Liao.

**Supervision:** Bei Xiao.

**Validation:** Chenxi Liao.

**Visualization:** Chenxi Liao, Masataka Sawayama, Bei Xiao.

**Writing – original draft:** Chenxi Liao, Bei Xiao.

**Writing – review & editing:** Chenxi Liao, Masataka Sawayama, Bei Xiao.

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
