## [Decision Letter · Decision Letter 0]

13 Jun 2024

Dear Mr Liao,

Thank you very much for submitting your manuscript "Probing the Link Between Vision and Language in Material Perception Using Psychophysics and Unsupervised Learning" for consideration at PLOS Computational Biology. As with all papers reviewed by the journal, your manuscript was reviewed by members of the editorial board and by several independent reviewers. The reviewers appreciated the attention to an important topic. Based on the reviews, we are likely to accept this manuscript for publication, providing that you modify the manuscript according to the review recommendations.

Sincerely,

Roland W. Fleming, PhD

Academic Editor

PLOS Computational Biology

Thomas Serre

Section Editor

PLOS Computational Biology

Reviewer's Responses to Questions

**Comments to the Authors:**

Reviewer #1: This paper presents a study about material appearance perception, comparing representations found by deep neural networks and human annotations for two different modalities: vision and language. For vision, the authors use image-based representations found by OpenCLIP and DINO models, and human visual similarity judgements (by placing images in a two-dimensional space); on the other hand, for language, they use semantic representations from various large language models (including OpenCLIP, but also Sentence-BERT or GPT2) and human descriptions of materials, in the form of natural language words. The objective is to analyze the vision-language relationship in the context of material perception, concluding that (within the conditions of their study), human visual judgements of material appearance correlate better with representations found by models that combine vision and language representations than with purely vision-based models. This conclusion, together with other interesting findings (e.g., highly lower correlations when using material names vs. removing them) is useful to design better computational models that mimic human visual perception.

Human perception of materials and how to model it with learning-based approaches is an open topic of research, and with the advent of large vision-language generative models (i.e., GPT, CLIP, diffusion...), I find it very important to thoroughly analyze the internal representations of such models and their correlations to human behavior, instead of using them as "black boxes". I think this paper does a very good job in this line, with useful contributions.

The paper is very easy to read, well motivated, and with a solid methodology. I find the approach convincing, and their design decisions reasonable (see below my only concerns in this regard).

I include here a couple of concerns that I would like the authors to better clarify/discuss in the paper:

* Regarding the dataset, I wonder why to limit their images to these specific three categories (soap, toy and rock) and their interpolations in the StyleGAN space. One could argue these images represent well a wide variety of material appearance (e.g., quite different optical and mechanical properties, including translucency), and the material morphing possibility could be very useful to analyze material properties "out of context". However, I would appreciate a better motivation of this selection of the data, and an explicit mention that their conclusions might be biased towards this particular space of appearance.

* Regarding the textual descriptions gathered from participants, the authors say that participants can "freely" articulate their experience with "unrestricted" descriptions. However, I have understood that the participants have a pre-defined set of characteristics to describe (e.g., material name, color) and they only use words or small phrases (vs. full sentences), so it sounds a bit overstated. I see it as a reasonable design decision, but it could better motivated and explained in the paper. Vision-language models, such as CLIP or GPT, can also manage long and free full-sentence descriptions, and the conclusions of the study (e.g., that participants are more diverse in the visual task than in the language one) could be different in this case. It would be helpful if the authors could offer more insights about this decision, or consider it as future work.

Also, a few questions and suggestions:

* The "Space of Morphable Material Appearance", up to what point is it "controllable"? I see great interpolation results, could we extrapolate also in this space?

* I find the "removing material names" experiment and analysis very interesting. Does it also exhibit a significant change in correlation for *morphed* materials without a clear category? Have you run any additional analysis on that?

* Did you consider including in the analysis the representations from a neural network trained *supervisedly* for a material similarity task?

* Typos / minor references corrections:

- l.86 behavior*al* tasks

- References consistency: et al. vs. list of authors

- Reimers et al. Sentence-BERT *EMNLP 2019*

- Deschaintre et al. The visual language of fabrics *ACM Trans. on Graphics 2023*

- Serrano et al. An intuitive color space... *ACM Trans. on Graphics 2016*

Reviewer #2: The project takes the excellent Soap-trained generative model from the authors' 2023 paper, and extends it by fine-training on additional photos in order to be able to generate images of Soap, Rock, "Squishy toys" (soft rubber/plastic/silicone), or any morph between these three categories, with variable lighting, colour, and background surface.

The work is very technically proficient, and the visual quality of the generated objects is exceptionally good. The continuous "Space of Morphable Materials" is an excellent stimulus-generation tool that has only in the past couple of years become technically possible. The creation of abstract object feature spaces that can be smoothly traversed, with visually photorealistic images at each point, but no physically-defined ground-truth, opens up lots of possibilities for probing perception.

Having generated a diverse set of 72 ambiguous-material stimuli, the authors then collect two sets of behavioural data: arrangements on a 2D screen in terms of visual similarity, and verbal descriptions which are input to a Large Language Model (of various types) in order to produce a measure of verbal similarity. These human-produced visual and verbal similarities are then compared in various ways: how well do visual and verbal similarities correspond within and across participants, as quantified by RDM correlation or an optimal transport algorithm? How well can visual features extracted by other DNNs predict the human visual similarity judgements? How about combining verbal similarities and DNN-computed image features to predict human visual similarities?

In each case the answer is broadly that the different similarity spaces correspond moderately well with one another, but don't explain all the non-noise variance. Figures 4B and 5 provide a nice intuition for the human data - in both visual and text-derived similarities, people create three broad categories, which overlap partially with object identity/generative model, and somewhat better with the perceived mechanical properties, preferred material name, and hue/saturation.

The main take-away message is that there's a fair bit of overlap between statistical image features, material properties, human-perceived visual similarity, and human-described material similarity (at least within this constrained material sub-space), but that verbal descriptions can't fully capture the image-to-image nuances we perceive.

These seem like unsurprising conclusions, but the project is a beautifully illustrated technical tour-de-force, and may serve as an inspiring proof-of-principle for a method to generate stimuli that may soon be widespread in psychology experiments.

MINOR COMMENTS

- It would be good to cite the very small number of other papers that have so far applied unsupervised learning to the study of material perception:

Metzger, A., & Toscani, M. (2022). Unsupervised learning of haptic material properties. _Elife_, _11_, e64876.

Storrs, K. R., Anderson, B. L., & Fleming, R. W. (2021). Unsupervised learning predicts human perception and misperception of gloss. _Nature Human Behaviour_, _5_(10), 1402-1417.

Reviewer #3: The authors investigate the link between visual judgments and verbal descriptions of material qualities. In a visual task they ask participants to arrange images of materials according to similarity. In a subsequent verbal task participants had to complete a verbal statement about materials qualities by filling in the blanks in a preformulated sentence. Data from both tasks was converted into Representational Dissimilarity Matrices and correlation as well as optimal transport between these RDMs assessed. At the individual level the correlation and optimal transport between visual and verbal RDMs were moderate, but increased at the group level. Deviations from ideal in the optimal transportation plan matrix were most frequently for within category stimuli and for mix-category stimuli. The authors conclude that visual and verbal representation align only coarsely, at the categorical level. And, that visual representations of materials are richer than verbalized semantic features.

Materials and their qualities play a huge role in everyday life, shaping our interactions with objects. Despite this important we still know very little material perception works, including how the brain represents materials. Therefore, to what extend verbal and visual representations of materials have a similar representational structure is a very interesting and important one. The authors use deep learning in combination with behavioral experiments to address this question. The proposed methods and modeling could prove useful beyond the area of material perception.

I have a few comments and questions that might help revising the manuscript.

Stimuli:

The stimuli were images of objects with different material qualities: soap, rock and squishy toy, as well as mixed version between these. Images were generated with a generative adversarial network that was first trained on real photographs of soap pieces and then subsequently on the other two categories. Mixed category stimuli were created by interpolating between layer-wise latent codes of respective trained GANS.

-Why were just 3 categories used, and why these particular ones? Looking at Figure 2 the differences between the material categories seems to be mostly shape related – optically, there seems to be quite a bit of overlap (even for the non-morphs, e.g. translucent ‘rocks’). What if the stimulus set was extended to one very different category (say wood or clay)

-Continuing the same argument: the main distinction between the three categories seems to be shape (not optical properties). Objects in all three categories are somewhat translucent. They differ, however in shape (it seems): e.g. soap is very regular, geometrically shaped (e.g. like platonic solids); rocks are irregular and jagged, and squishy toys all seem rounded. It would have been interesting to add this 3Dshape aspect as an answer option to the verbal description and to check how much this factor contributes to visual similarity judgements.

Procedure:

All participants first completed the Multiple Arrangement task, and then the Verbal Description task.

– could this cause an order effect? Why not counterbalancing the order?

Verbal responses: ‘We used a fixed template to concatenate the five aspects that participants described an image: “It is a material of [material name] with the color of [color], it is [optical properties], it is [mechanical properties], and it is [surface texture].”

-In a sense this template forces the participants’ responses to a certain level of coarseness. Essentially the template encouraged participants to fill in the blanks with single or multiple adjectives. However, one can describe the look and potential feel of materials stylistically much more diverse than just listing a bunch of adjectives. I can imagine that this format was used because it is more suitable to use with LLMs, however it should be acknowledged that the potential complexity of the verbal response was restricted and – that this might have shaped the findings, e.g. ‘the persistent gap in using words to capture the nuanced visual differences among diverse material samples’, or the higher interparticipant correlation in the verbal task.

-In general, I think what is communicable about an image/material certainly depends on the linguistic skill of the communicator

Figure 5: Displays the most frequently used word for each image aggregated across all participants.

-Was there a cut off for minimum frequency? Suppose if all but two participants used a different label, than that label (used by 2 participants) would be the most frequent, even though it was not used very often overall.

-In the verbal task, do the authors think that being asked to provide a material name might bias/limit the subsequent description of the attributes? E.g. if I label this rock then it must be rough.

-Figure 6 Participant B, who has the highest number of unique words also has lowest correlation between visual and verbal RDMs. Is a (relatively) high number of different words possibly “punished” by this analysis, because there will be less correspondence between any two stimuli (descriptions)?

-How comparable in terms of size are the feature vectors for visual and verbal representations? Perhaps this could be added to the methods.

Discussion page Line 263 ff: ‘Our results show that when the material name is removed from the text embedding, the correlation between Vision and Text RDMs systematically decreases across participants (Figure 6). This may stem from the functional roles that nouns (material play in everyday language usage56 and highlights the significant role of material names in visual categorization. ‘

-This sounds a bit vague. Can the authors explain how removing the material name changes the representation/weights in the LLM? Can this be probed?

**Have the authors made all data and (if applicable) computational code underlying the findings in their manuscript fully available?**

Reviewer #1: **No: **I confirm that some data and code to do the cross-material morphing is already available in the provided Github repository, but not all (e.g., trained networks or data analysis code are missing). The authors have promised to upload all upon publication.

Reviewer #2: Yes

Reviewer #3: Yes

PLOS authors have the option to publish the peer review history of their article (what does this mean?). If published, this will include your full peer review and any attached files.

Reviewer #1: No

Reviewer #2: No

Reviewer #3: No

Figure Files:

Data Requirements:

Reproducibility:

References:

---

## [Decision Letter · Decision Letter 1]

11 Sep 2024

Dear Mr Liao,

We are pleased to inform you that your manuscript 'Probing the Link Between Vision and Language in Material Perception Using Psychophysics and Unsupervised Learning' has been provisionally accepted for publication in PLOS Computational Biology.

Best regards,

Roland W. Fleming, PhD

Academic Editor

PLOS Computational Biology

Thomas Serre

Section Editor

PLOS Computational Biology

Reviewer's Responses to Questions

**Comments to the Authors:**

Reviewer #1: The authors convincingly addressed the reviews' concerns and I think it is a solid paper with high interest to the research community in material appearance.

Reviewer #3: Thank you for the careful revisions, additions and answers to my questions. The ms looks great.

**Have the authors made all data and (if applicable) computational code underlying the findings in their manuscript fully available?**

Reviewer #1: **No: **Not yet, they have promised to do so upon publication

Reviewer #3: Yes

PLOS authors have the option to publish the peer review history of their article (what does this mean?). If published, this will include your full peer review and any attached files.

Reviewer #1: No

Reviewer #3: No

---

## [Editor Report · Acceptance letter]

27 Sep 2024

PCOMPBIOL-D-24-00728R1 

Probing the Link Between Vision and Language in Material Perception Using Psychophysics and Unsupervised Learning

Dear Dr Liao,

I am pleased to inform you that your manuscript has been formally accepted for publication in PLOS Computational Biology. Your manuscript is now with our production department and you will be notified of the publication date in due course.

With kind regards,

Jazmin Toth
